∂ | **Open Peer Review** | Host-Microbial Interactions | Research Article

# The plant pathogenic bacterium *Candidatus* Liberibacter solanacearum induces calcium-regulated autophagy in midgut cells of its insect vector *Bactericera trigonica*

**Poulami Sarkar,**[1] **Ola Jassar,**[1,2] **Murad Ghanim**[1]

**ABSTRACT** Autophagy plays an important role against pathogen infection in many organisms; however, little has been done with regard to vector-borne plant and animal pathogens, that sometimes replicate and cause deleterious effects in their vectors. *Candidatus* Liberibacter solanacearum (CLso) is a fastidious gram-negative phloem-restricted plant pathogen and vectored by the carrot psyllid, *Bactericera trigonica*. The plant disease caused by this bacterium is called carrot yellows and has recently gained much importance due to worldwide excessive economical losses. Here, we demonstrate that calcium ATPase, cytosolic calcium, and most importantly Beclin-1 have a role in regulating autophagy and its association with Liberibacter inside the psyllid. The presence of CLso generates reactive oxygen species and induces the expression of detoxification enzymes in the psyllid midguts, a main site for bacteria transmission. CLso also induces the expression of both sarco/endoplasmic reticulum Ca2+pump (SERCA) and 1,4,5-trisphosphate receptors (ITPR) in midguts, resulting in high levels of calcium in the cellular cytosol. Silencing these genes individually disrupted the calcium levels in the cytosol and resulted in direct effects on autophagy and subsequently on Liberibacter persistence and transmission. Inhibiting Beclin1-phosphorylation through different calcium-induced kinases altered the expression of autophagy and CLso titers and persistence. Based on our results obtained from the midgut, we suggest the existence of a direct correlation between cytosolic calcium levels, autophagy, and CLso persistence and transmission by the carrot psyllid.

**IMPORTANCE** Plant diseases caused by vector-borne Liberibacter species are responsible for the most important economic losses in many agricultural sectors. Preventing these diseases relies mostly on chemical sprays against the insect vectors. Knowledge-based interference with the bacteria-vector interaction remains a promising approach as a sustainable solution. For unravelling how Liberibacter exploits molecular pathways in its insect vector for transmission, here, we show that the bacterium manipulates calcium levels on both sides of the endoplasmic reticulum membrane, resulting in manipulating autophagy. Silencing genes associated with these pathways disrupted the calcium levels in the cytosol and resulted in direct effects on autophagy and Liberibacter transmission. These results demonstrate major pathways that could be exploited for manipulating and controlling the disease transmission.

**KEYWORDS** autophagy, Liberibacter, psyllid, calcium signaling, SERCA

Autophagy is an evolutionary cellular process of recycling intercellular components and maintaining cellular functions (1). It plays a protective role during endoplasmic reticulum (ER) stress for protecting the cells from metabolic damage and is essential for cellular homeostasis and development (1). Autophagy processes initiate with the

Address correspondence to Murad Ghanim, ghanim@volcani.agri.gov.il.

The authors declare no conflict of interest.

See the funding table on p. 12.

formation of double-membrane vesicles known as autophagosomes (APs) from the ER, involving vesicular engulfment of materials and pathogens or foreign particles, and fusion with lysosomes for lysosomal hydrolases and catabolic processes (2, 3). This in turn reduces ER stress which consequently regulates apoptosis. This coordination between autophagy and apoptosis is crucial for regulating cell survival or cell-death, respectively (4–6). High cytosolic calcium ($Ca^{2+}$) during ER stress is one of the multiple signaling molecules regulating the induction of autophagy (7, 8). 1,4,5-trisphosphate receptors (ITPRs/IP3Rs) are tetrameric $Ca^{2+}$ channels located at ER, which release $Ca^{2+}$ from the ER to the cytosol and cytosolic $Ca^{2+}$ re-enters the ER through a $Ca^{2+}$ pump called ATP2A/SERCA (sarco/endoplasmic reticulum $Ca^{2+}$) (9–13).

Cytosolic $Ca^{2+}$ plays an important role as a pro-autophagic signal encompassing both Beclin1 and mTOR (mechanistic target of rapamycin) signaling cascades (14, 15). Beclin1, an ortholog of yeast Atg6, represents a determining link between autophagy and apoptosis and is crucial for the initiation of autophagosome formation. Beclin1 when bound to Bcl2 inhibits autophagy and induces mitochondrial apoptosis. This interaction is dynamically regulated by death-associated protein kinase (DAPK), which phosphorylates Beclin1 at Thr119 antagonizing the interaction, thereby inducing autophagy (16–18). Serine/threonine kinase mTOR, in particular complex1 (mTORC1), is the master negative regulator of autophagy, which is inhibited by AMPK (AMP-activated protein kinase) which in turn is activated by $Ca^{2+}$-calmodulin-dependent protein kinase kinase-β (CaMKKβ) upon increase in cytosolic $Ca^{2+}$ (Ca2+-CAMKK2-AMPK pathway). AMPK also regulates autophagy by phosphorylating Beclin1 at Thr388 (18, 19). Active mTORC1 triggers inactivation of ULK (Unc51-like kinase, homolog of ATG1), a serine/threonine kinase that plays a key role in inducing autophagy (20–22). Loss of mTORC1 activity induces activated ULK to initiate autophagy by phosphorylating Beclin1 at Ser15 (18). Hence, Beclin1 accounts for the core molecular machinery involved in autophagy.

Bacterial pathogens often induce ER stress and free cytosolic $Ca^{2+}$ in the host cells which in consequence induces autophagy as a defense mechanism to cellular stress (23, 24). However, many pathogenic bacteria can manipulate the autophagic pathway and replicate inside the autophagosomal compartment (23, 25, 26). Psyllid-Liberibacter relationship is one such system where the molecular mechanism of interaction and pathogenesis is obscure. One of the Liberibacter species, *Candidatus* Liberibacter solanacearum (CLso), Haplotype D is a known gram negative, yet unculturable bacterium transmitted by the carrot psyllid, *Bactericera trigonica* in a circulative and persistent manner. Recent molecular and transcriptomic analyses revealed induced ER stress and autophagic genes in psyllids in response to Liberibacter (27–30). Hijacking host immunity is crucial for bacterial survival and replication inside the host cells, and autophagy seems important to reduce bacteria-induced cellular ER stress and maintain homeostasis. In recent studies, CLso was reported to induce autophagy in the psyllid host, *B. trigonica* (31) and in *Diaphorina citri* (30), and repress apoptosis in *Bactericera cockerelli* (32).

In this study, we investigated the role of $Ca^{2+}$ signaling and autophagy in CLso propagation inside its psyllid vector. In addition, we also studied the role of phosphorylated Beclin1 in autophagy initiation. We found elevated cytosolic $Ca^{2+}$ and increase in autophagy-regulated genes in the psyllid midguts in response to CLso infection. Blocking the $Ca^{2+}$ channels and the protein kinases in the calcium-signaling cascade drastically disturbed both Beclin1 regulation, leading to alterations in autophagy and CLso levels.

## RESULTS

### CLso induces ROS generation in psyllid midguts

Psyllid midguts showed an increased expression of reactive oxygen species (ROS; Fig. 1A through C) with enhanced expression of cytochrome P450 (C450) and the detoxification enzyme superoxide-dismutase (SOD) both in midguts and whole insects. However, there was a decline in the expression of glutathione S-transferase (GST; Fig. 1B and

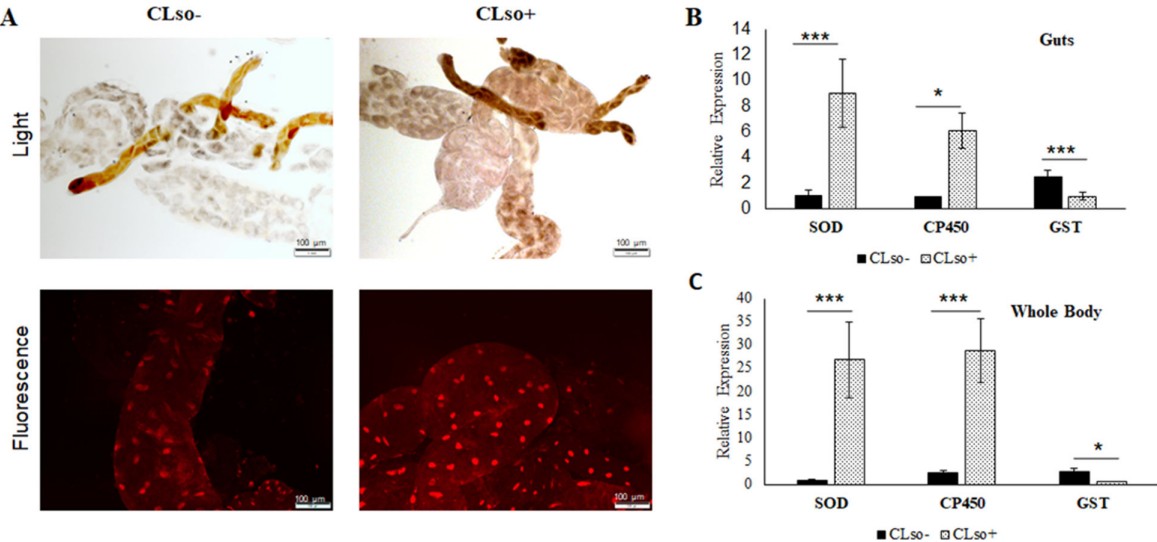

**FIG 1** ROS detection in CLso-infected psyllid midguts. (A) Light and fluorescence detection of ROS using DHE stain. (B) Real-time PCR analysis for SOD, CP450, and GST in guts and (C) in whole psyllids. * indicates $P \leq 0.05$ and *** indicate $P \leq 0.01$. Error bars denote SE with $n \geq 10$.

C). Increased ROS was observed in CLso+ psyllid midgut nuclei (bright fluorescent red) upon dihydroethidium (DHE) intercalation in response to higher oxidation (Fig. 1A). The intensities of fluorescence were measured with ImageJ (Fig. S1).

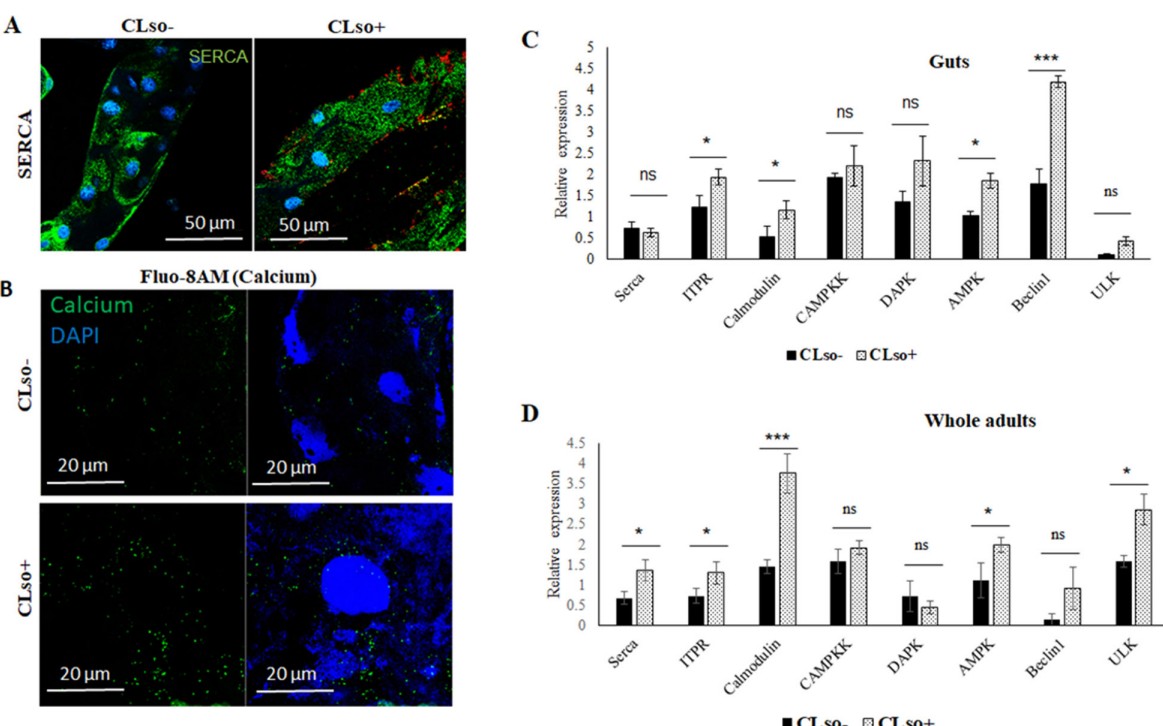

**FIG 2** Differential expressions of SERCA, calcium, and calcium-signaling genes. (A) Immunostaining of SERCA (green) and CLso (red) in CLso-free (CLso–) and CLso-infected (CLso+) psyllid midguts counterstained with DAPI (blue). (B) Detection of cytosolic calcium levels (green) using Fluo-8AM staining in CLso– and CLso+ psyllid midguts, counterstained with DAPI (blue). (C) Real-time PCR analysis for the expression of SERCA, ITPR, and calcium-signaling cascade genes in the midguts and (D) in whole psyllids. * indicates $P \leq 0.05$, *** indicates $P \leq 0.01$, and ns indicates not significant. Error bars denote SE with $n \geq 10$.

## Induced expression of calcium ATPases and calcium signaling genes with increased cytosolic calcium levels in CLso-infected psyllids

*In situ* expression of SERCA (sarco/endoplasmic reticulum $Ca^{2+}$), which is responsible for calcium influx from the cytosol to the ER, was observed using immunolocalization with SERCA antibody. Expression of SERCA was highly induced in CLso+ psyllid midguts (Fig. 2A), where it was observed around the nuclei stained with DAPI (blue), a reminiscent of ER staining. Higher amounts of calcium were also detected in the CLso+ midgut cytosol when stained with calcium staining fluophore Fluo-8AM (Fig. 2B). Real-time PCR expression analysis revealed upregulation of SERCA as well as some of the calcium-induced genes in the calcium signaling pathway that result in the induction of autophagy (Fig. 2B and C). The intensities of the signal were measured and verified the expression and staining results (Fig. S2A and B).

## Inhibition of the calcium pumps in the ER membrane alters CLso titer

To study the role of calcium ATPases and calcium in influencing CLso levels, we silenced SERCA (responsible for calcium influx) and ITPR (responsible for calcium efflux) individually with double-stranded RNA (dsRNA) as described in the methods section. dsSERCA-treated CLso+ psyllids showed reduction in SERCA gene expression in the midguts in both real-time and immunolocalization experiments (Fig. 3A and B). This resulted in increased accumulation of cytosolic calcium in the midguts as analyzed using calcium binding fluophore, Fluo-8AM (Fig. 3C). Moreover, CLso titers drastically reduced as a result of silenced SERCA and consequent high-calcium levels in the cytosol (Fig. 3A and B). The intensities of the signals were quantified using ImageJ and confirmed the obtained results (Fig. S2C and D). Lysotracker was used to verify the increase in auto-

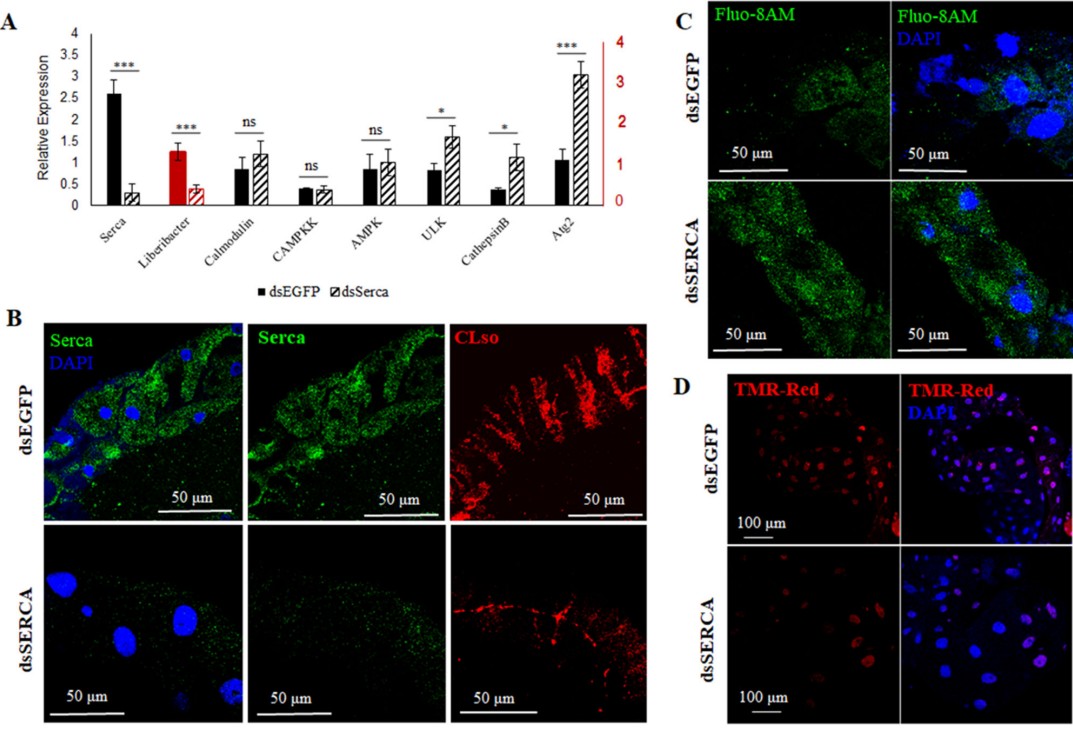

FIG 3  Effect of silencing SERCA on CLso and autophagy. (A) Real-time PCR analysis of the differential expression change in SERCA and corresponding calcium-signaling genes along with CLso abundance in the psyllid midguts following SERCA silencing. * denotes $P \leq 0.05$, *** indicates $P \leq 0.01$, and ns indicates not significant. Error bars denote SE with $n \geq 10$. (B) Representative image of immunostaining analysis of SERCA (green) and CLso (red) in the midguts upon SERCA silencing, counterstained with DAPI (blue). (C) Elevated levels of cytosolic calcium (green) after silencing SERCA in midguts. (D) TUNEL assay showing reduced apoptosis in SERCA silenced midguts.

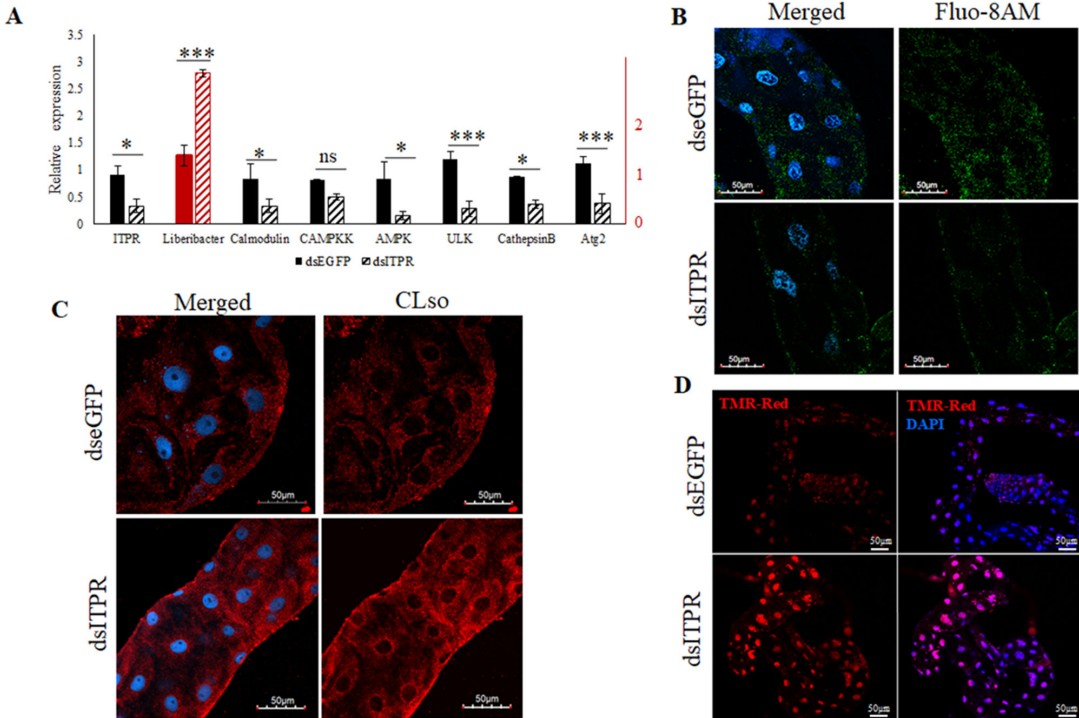

**FIG 4** Effect of ITPR silencing on CLso and autophagy. (A) Real-time PCR analysis showing differential expression levels of ITPR, calcium-signaling genes, and CLso abundance in the psyllid midguts following silencing ITPR. * denotes $P \leq 0.05$, *** denotes $P \leq 0.01$, and ns indicates not significant. Error bars denote SE with $n \geq 10$. (B) Fluo-8AM staining of cytosolic calcium (green) after ITPR silencing and counterstaining with DAPI (blue). (C) Immunostaining of CLso (red) in the psyllid midguts following ITPR Silencing. (D) Increased apoptosis in ITPR-silenced midguts as detected by TUNEL assay using TMR-red (red) and DAPI (blue).

lysosomes in the psyllid midguts (Fig. S3). This also resulted in decreased apoptosis as visualized with TMR-Red staining (Fig. 3D).

On the contrary, dsITPR silenced midguts showed a reduction in cytoplasmic calcium levels and reduction in corresponding genes involved in the calcium signaling pathway along with increased CLso abundance in the midguts (Fig. 4A and C). The intensities of the signals for CLso were also analyzed using ImageJ (Fig. S2E and F). The decrease in autophagy was also verified using Lysotracker staining (Fig. S3). Moreover, when the CLso+ psyllid midguts were stained with TMR-Red, there was an increase in apoptosis upon silencing ITPR expression (Fig. 4D).

In addition, Liberibacter was observed to localize at the surface of the midgut cells in a stripe-like pattern at one focal plane and also around the nuclei as the focal plane is changed (Fig. S4).

## Inhibition of Beclin1 signaling cascade

Beclin1 phosphorylation was reduced *in vivo* by chemically inhibiting AMPK and DAPK, which phosphorylates Beclin1 at sites Ser93, Ser96, and Thr119, respectively. Phosphorylation of the sites Thr119, Ser93, and Ser96 was observed to be more in the CLso-infected psyllid midguts than in CLso-free midguts (Fig. 5 and 6). The intensity measurements of the signals were also calculated using ImageJ (Fig. S2G and H). As expected, Beclin1 phosphorylation at sites Ser93, Ser96 was drastically reduced when AMPK was inhibited. This consequently resulted in increase in CLso abundance in the midguts. We also tested the effect of this inhibition on autophagy, and the autophagy-related genes, specifically Atg2 which was significantly downregulated (Fig. 5). Additionally, the presence of lysosomes was reduced (Fig. 5C) along with autophagic vacuoles (Fig. S5) which indicated less autophagy. In contrast, for DAPK inhibition, there were no significant changes in CLso abundance even when Beclin1 remained un-phosphorylated at site

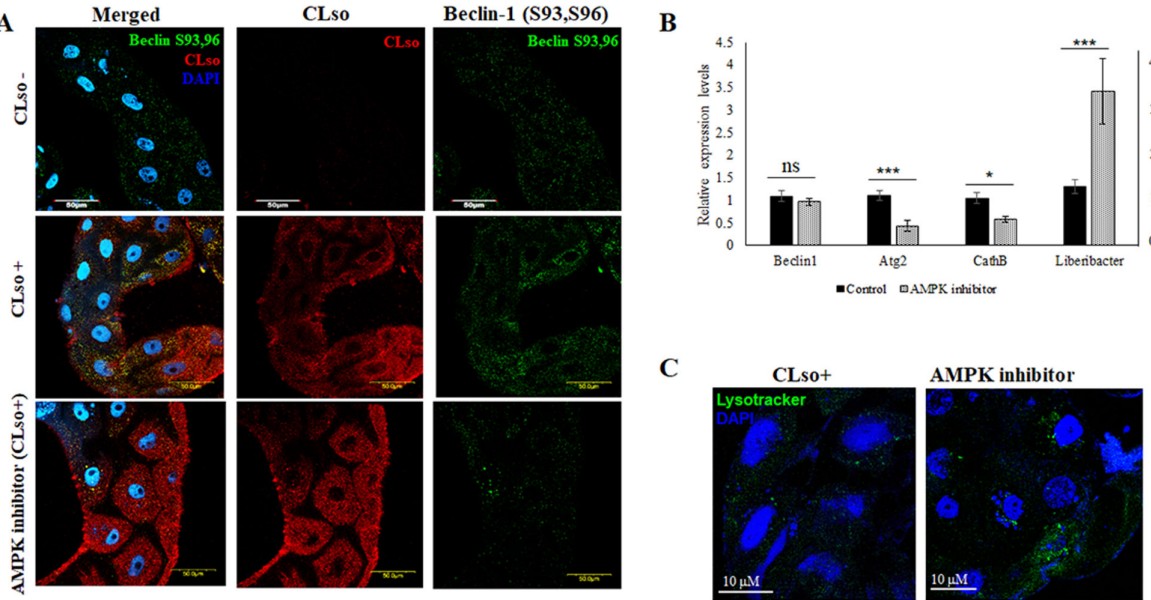

**FIG 5** Effect of AMPK inhibitor on Beclin1 phosphorylation, autophagy, and CLso. (A) Immunostaining analysis reveals reduced Beclin1 phosphorylation (green) at Ser93,96 sites and higher accumulation of CLso (red) in the AMPK-inhibited psyllid midguts, counterstained with DAPI (blue). (B) Real-time PCR assay for the differential expression of Beclin1, autophagy genes, and CLso abundance following AMPK inhibition. $P \leq 0.05$ is indicated by *, $P \leq 0.01$ by ***, and ns indicates not significant. Error bars denote SE with $n \geq 15$. (C) Autolysosome detection using Lysotracker DND (green) counterstained with DAPI (blue).

Thr119 (Fig. 6A and B). There was no significant difference in Atg2 gene expression; however, there was a slight increase in the formation of autophagic vacuoles in the midguts as seen with monodansylcadaverine (MDC) staining, following DAPK inhibition (Fig. 6B and C; Fig. S5).

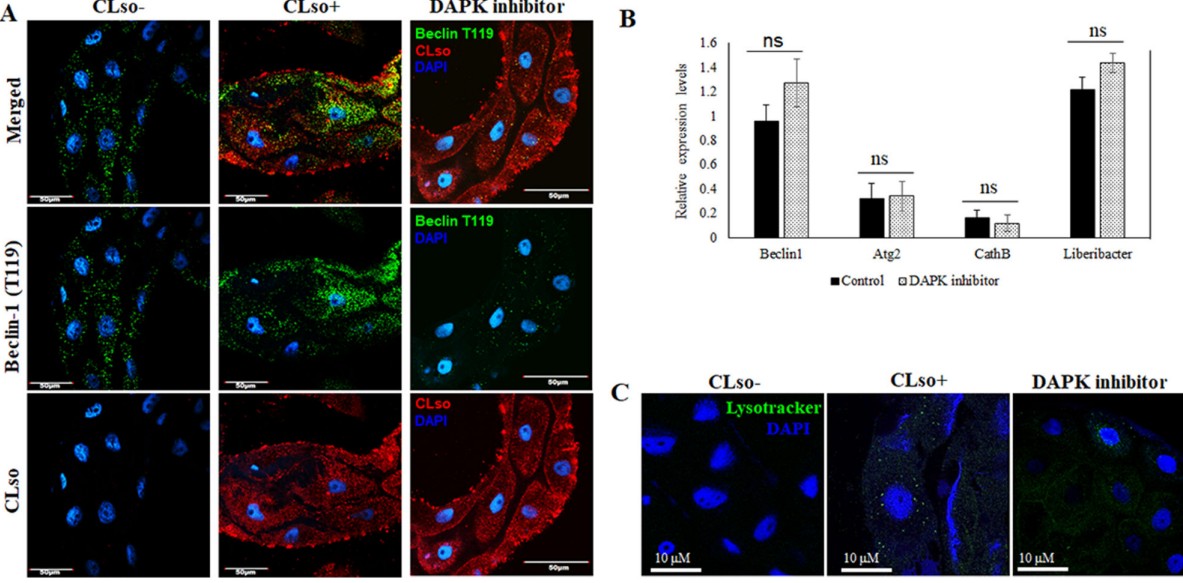

**FIG 6** Effect of DAPK inhibitor on Beclin phosphorylation, autophagy and CLso abundance. (A) Immunostaining analysis for Beclin1 phosphorylation at site Thr119 (green) and CLso (red) counterstained with DAPI (blue). (B) Real-time analysis showing differential expression levels for Beclin1, autophagy genes and CLso titer. ns denotes not significant. Error bars denote SE with $n \geq 15$. (C) Autolysosome detection using Lysotracker DND (green) counterstained with DAPI (blue).

## DISCUSSION

CLso, Haplotype D is an intracellular, gram-negative bacterium transmitted by the carrot psyllid, *B. trigonica*. Understanding the stress response and innate immunity of the vector is crucial to determine new approaches for disrupting the disease transmission. Innate immunity involves both autophagy and apoptosis and functions to maintain homeostasis during pathogenesis (4, 33, 34). Autophagy maintains cellular homeostasis during ER stress by eliminating intracellular pathogens and delivering them to lysosomes for destruction (2, 3, 35, 36). One of the major signaling pathways that regulate autophagy is calcium (Ca2+) ions that signal Beclin1 through different protein kinases present in the calcium signaling cascade, leading to initiating autophagy (7, 8, 14, 15). A number of recent studies involving Liberibacter species have demonstrated the activation of autophagy in insect vectors during pathogen infection (30, 31). CLso induces the formation of a double-membrane autophagic vacuoles from the ER membrane (30). In addition, major autophagy-related genes are reported to be upregulated in Asian citrus psyllid (30) and the potato psyllid (37) following Liberibacter infection.

In the current study, we attempted to explore the role of calcium and the calcium-signaling cascade proteins involved in autophagy in the carrot psyllids during CLso infection. As pathogenesis induces ROS, which in turn stimulates autophagy (38), we also analyzed ROS formation in the psyllid midgut. DHE staining of the midguts showed higher ROS levels in CLso-infected psyllids as compared with CLso-free psyllids (Fig. 1). Real-time expression studies of detoxification enzymes showed upregulation of SOD and cytochrome P450 genes in CLso-infected psyllids with downregulation of GST (Fig. 1). Next, we tested the expression of $Ca^{2+}$ ATPases and the cytosolic $Ca^{2+}$ levels in the psyllid midguts upon CLso infection. Immunostaining of SERCA revealed higher intensity of signal in CLso-infected midguts (Fig. 2), which also showed higher signals for cytosolic calcium (Fig. 2). Expression of $Ca^{2+}$ influx (SERCA) pumps were elevated in CLso-infected midguts along with other protein kinases involved in the calcium signaling pathway including Beclin1 (Fig. 2). This elevation in SERCA could be a mechanism for maintaining calcium homeostasis and reducing ER stress caused by CLso. However, it is interesting to observe high levels of cytosolic calcium despite overexpression of SERCA. We tested the expression of $Ca^{2+}$ efflux pump protein, ITPR which was also found to be overexpressed in the CLso-infected psyllids (Fig. 2).

To understand the involvement of these two pumps in CLso propagation and autophagy, we next silenced both SERCA and ITPR individually using dsRNA. Silencing SERCA was validated using both real-time expression analysis and immunostaining (Fig. 3). Silencing SERCA showed increased levels of cytosolic calcium in the midgut (Fig. 3) as well as elevated levels of calcium signaling genes, autophagy-related gene-2 (Atg2), and Cathepsin-B (CathB) (Fig. 3). Atg2 is one of the potential linking proteins between ER and autophagophores and mediates denovo autophagosome biogenesis (39), whereas CathB is an essential lysosomal hydrolase (40, 41). The sequences for Atg2 and CathB were available from the denovo transcriptome analysis (29). Silencing SERCA also resulted in a higher number of autolysosomes (Fig. S3) and reduction in apoptosis (Fig. 3). As it is known that autophagy and apoptosis cross-regulate each other (4–6), our results verified upregulation in autophagy upon elevated cytosolic $Ca^{2+}$ levels caused by silencing SERCA. Consequently, there was a reduction in the levels of CLso in dsSERCA-treated midguts (Fig. 3). Comparably, when ITPR was silenced using dsRNA, the calcium-signaling genes and autophagy-related gene 2 were downregulated (Fig. 4) along with reduced cytosolic $Ca^{2+}$ levels (Fig. 4). This resulted in increased levels of CLso in the dsITPR treated midguts (Fig. 4). In general, caspase-3 cleavage assays are strong indicators of apoptosis; however, we lacked sequence data information for the carrot psyllids, and we carried out a corresponding LysoTracker and TUNEL assay which confirmed higher apoptosis (Fig. 4) and reduction in autophagy (Fig. S3) in the dsITPR-treated midguts.

As Beclin1 is crucial for autophagy and results in initiating autophagosome formation, we wanted to examine the role of two important protein kinases involved in the calcium-signaling pathway in Beclin1-mediated autophagy. As depicted in Fig. 7,

cytosolic Ca$^{2+}$ activates both DAPK and AMPK, which in turn phosphorylates Beclin1 at specific sites to initiate autophagosome formation. In this study, we tried to chemically inactivate both DAPK and AMPK individually, for testing which protein kinase is crucial for Beclin1-mediated autophagy. AMPK inhibitor drastically reduced the phosphorylation of Beclin1 at S93,96 sites, leading to accumulation of CLso in the treated midguts (Fig. 5). This also led to reduction in autophagy as analyzed using real-time PCR expression analysis, Lysotracker (Fig. 5) and MDC-autophagic vacuole staining (Fig. S5). On the other hand, DAPK inhibitor reduced the phosphorylation of Beclin1 at site Thr119 (Fig. 6); however, there was no significant change in CLso abundance or in autophagy-related genes (Fig. 6). Interestingly, there was a slight increase in the formation of autophagic vacuoles as stained with MDC (Fig. S5). This suggests that Beclin-mediated autophagy is AMPK dependent, and DAPK inhibition does not induce autophagy (with no lysosomal activity) even if it leads to a slight increase in autophagic vacuole formation. It can be said that CLso-mediated autophagy is independent of DAPK phosphorylation. Further analyses are required to identify the crosstalk between autophagy and apoptosis using pathway-specific markers that determine the role of Beclin. Mitochondrial membrane potential-dependent dyes, caspase cleavage assays, and Cytochrome C release assays would validate the occurrence of apoptosis whereas detection of LC3 proteins (not detected in carrot psyllid transcriptome) would determine the autophagy process.

In conclusion, this study demonstrated that cytosolic Ca$^{2+}$ regulates autophagy through activating AMPK. It also revealed that AMPK is crucial for Beclin1 role in initiating autophagy, and autophagy and CLso propagation negatively regulate each other to maintain a balance between the pathogen and the vector. In future studies, it will be interesting to explore the exact mechanism leading to elevated calcium levels in the cytosol and to test whether autophagy responds to Liberibacter as a part of the innate immunity of the insect.

## MATERIALS AND METHODS

### Insects and plants material used

*Carrot* psyllids were collected from a carrot field in Saad, Israel and were maintained for generations in the laboratory. *Ca*. Liberibacter solanacearum - infected (CLso+) and *Ca*. Liberibacter solanacearum - free (CLso-) maintained on parsley (*Petroselinum crispum*) in separate cages under 14 h photoperiodic light, 25±2°C and 60% humidity. Psyllids were periodically tested for CLso using PCR analyses (Primers: Omp_F/Omp_R) (42).

### ROS imaging

*In situ* ROS detection was carried out using DHE (Sigma-Aldrich, Israel), which intercalates into nucleic acid. Briefly, midguts were dissected out of psyllids in PBS (phosphate buffer saline) and immediately incubated in 10 mM DHE for 7 min in the dark. The midguts were washed with PBS thrice before mounting in glass slides and viewed under both light microscope and confocal microscope at excitation of 535 nm and emission of 610 nm.

### dsRNA preparation and treatment

dsRNA for SERCA (dsSERCA-CL1868.Contig2_DeNovo) and ITPR (dsITPR- Unigene8082_DeNovo) and control dsRNA (ds-eGFP-MK387175.1) was prepared using the T7 FlashScribe transcription kit (Cellscript, USA). PCR-amplified products using gene-specific primers containing T7 promoters on 5′ end was used for dsRNA production (Table 1). dsRNA quality and quantity were checked using agarose gel electrophoresis and NanoDrop 1,000 spectrophotometer (Thermoscientific). dsRNA feeding was carried out using fresh leaf flush as previously described (42). ds-eGFP was used as a control. The silencing efficiency was checked by both qRT-PCR and protein-specific antibody. Each experiment was replicated thrice with minimum of 10 insects for each treatment.

**TABLE 1** Primers used in this study[a]

| Primer name | Sequence 5'>3' | Target gene | Product size (bp) | Reference |
|---|---|---|---|---|
| **Primers used for qPCR** | | | | |
| CaATPase_Fq | AGACAAGATCCCGGCTGAC | *B. trigonica* SERCA | 176 | This study |
| CaATPase_Rq | CCACGTTGGTTCCGGAGAA | | | |
| ITPR_Fq | CAGTACTCTGGGCCTTGTG | *B. trigonica* ITPR | 156 | This study |
| ITPR_Rq | TCTGTTTGGCAGCTTTCCAG | | | |
| ULK_Fq | GATTTCGGCTTCGCTGAGTT | *B. trigonica* ULK | 156 | This study |
| ULK_Rq | AAACAGGGCCTCGTATGTGA | | | |
| Calmodulin_Fq | TGCTCGTCCGTAGGTTTCTT | *B. trigonica* Calmodulin | 132 | This study |
| Calmodulin_Rq | GAATTACTTGGGCAGTGACG | | | |
| CAMPKK_Fq | ACATCAAGCCGTCCAACCTA | *B. trigonica* CAMPKK | 184 | This study |
| CAMPKK_Rq | TCATAGGCCTTGCCACTGAA | | | |
| DAPK_Fq | AGCTCTGGCAACATTGAGGA | *B. trigonica* DAPK | 174 | This study |
| DAPK_Rq | CACCTCATGCAGGATTTCC | | | |
| AMPK_Fq | AATGGGAAGGAGGCGGTAAA | *B. trigonica* AMPK | 126 | This study |
| AMPK_Rq | ACTCATGCTCTCCTTCAGGT | | | |
| Beclin1_Fq | CTTCTCTCCTCTCCTCGC | *B. trigonica* | 173 | This study |
| Beclin1_Rq | GCTACATAGGCACGGGCAA | Beclin1 | | |
| Atg2F | TGTGGCCCAGTGTGTCATT | *B. trigonica* | 147 | This study |
| Atg2R | ACTGTTGCCTGTCTTGCCC | Atg2 | | |
| CathBF | CAAGTCTGGTGTGTACAAGCA | *B. trigonica* | 125 | This study |
| CathBR | TGTTCCACGAATTGGCGATC | CathepsinB | | |
| qOmp_F | ATGCCACGTGAAGGTTTGAT | CLso Outer membrane | 152 | (43) |
| qOmp_R | AGATGACCCAGATCATGTTTGA | proteinA | | |
| psy_GST_F | AATGGAAAGCTTTCGTGGGC | *B. trigonica* | 177 | This study |
| psy_GST_R | TTAATTTTCAGTCACTGGTCTTTTTG | Glutathione S-Transferase | | |
| psy_SOD_F | AACAATATCATCGGCAGAACG | *B. trigonica* | 138 | This study |
| psy_SOD_R | TCATGCCTTGGTGATGCCAA | Superoxide dismutase | | |
| psy_Cp450_F | GGTGACAAGTCAGTGCTACG | *B. trigonica* | 121 | This study |
| psy_Cp450_R | ACATGAACGTGTCCACCTCT | Cytochrome P450 | | |
| **Primers used for dsRNA** | | | | |
| ds_CaATPase_F | TAATACGACTCACTATAGGGCTTCGTGGAGCCGTTTGTG | *B. trigonica* SERCA | 432 | This study |
| ds_CaATPase_R | TAATACGACTCACTATAGGGCCACGTTGGTTCCGGAGAA | | | |
| ds_ITPR_F | TAATACGACTCACTATAGGGTCAGTACTCTGGGCCTTGTG | *B. trigonica* ITPR | 383 | This study |
| ds_ITPR_R | TAATACGACTCACTATAGGGACCTCAGCTTGTAGAACGG | | | |
| ds_eGFP_F | TAATACGACTCACTATAGGGTTCATCTGCACCACCGGC | eGFP sequence | 464 | (42) |
| ds_eGFP_R | TAATACGACTCACTATAGGGTAGTGGTTATCCGGGAGGA | | | |

[a]The primers designed in this study are derived from the sequences of the de novo reads in the psyllid transcriptome (29).

## DAPK and AMPK inhibition

To identify which pathway is most important for Beclin1-mediated autophagy, Beclin1-phosphorylation by DAPK and AMPK was inhibited using DAPK inhibitor and AMPK inhibitor (Merck), respectively. 10 µg/mL of inhibitor in ethanol was applied on a fresh parsley flush with around 20 psyllids in a jar. After 16 h, the insects were collected for DNA/RNA isolation or dissection for microscopy. Ethanol was applied to a flush as a control experiment. Each experiment was replicated thrice with a minimum of 15 insects for each treatment. Beclin1 phosphorylation corresponding to each treatment was detected using antibodies as described later in this study.

## Calcium, lysotracker, and MDC staining

To detect the cytosolic calcium, we used Fluo-8AM (Abcam). Briefly, the midguts were dissected in PBS and incubated with Fluo-8AM in HBSS (Hank's balanced salt solution) for 30 min for 1 h in 37°C in dark. The midguts were then washed with PBS thrice and was

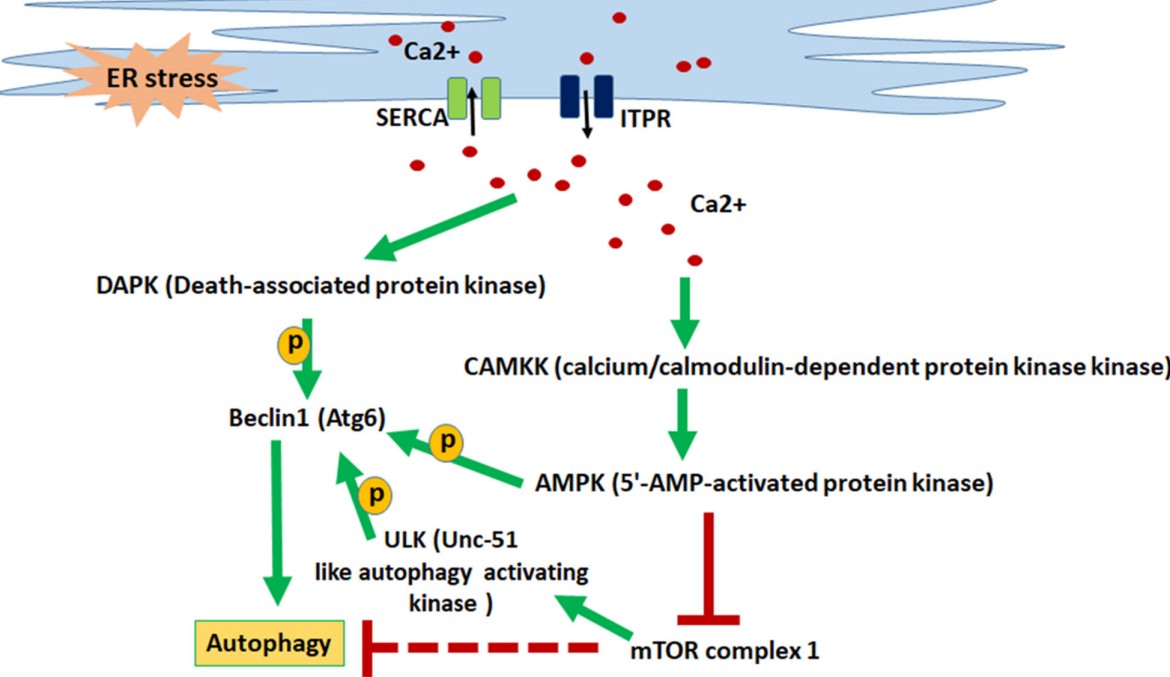

**FIG 7** Depiction of calcium signaling cascade leading to autophagy. Calcium influx (SERCA) pumps and efflux (ITPR) pumps maintain calcium homeostasis in the ER as well as in the cytosol. ER stress causes elevated cytosolic calcium levels, which in turn activates downstream protein kinases leading to the activation of Beclin1 which helps in the initiation of autophagosome formation leading to autophagy.

mounted with DAPI for microscopy. Autolysosomes were detected using LysoTracker Green DND-26 (Invitrogen) as described previously (31). In addition, autophagic vacuoles were labeled by MDC (Sigma Aldrich, Israel) stain. Briefly, the guts were dissected in PBS and stained in 4 mM MDC for 90 min in dark. The guts were then washed with PBS and were mount with DAPI to observe under the confocal microscope in green channel (emission: 525 nm and excitation: 488 nm).

## Terminal deoxynucleotidyl transferase dUTP nick end labeling

The psyllid midguts were diagnosed for cell-death or apoptosis using *in situ* cell-detection kit TMR-red (Roche) following the manufacturer's instructions. Briefly, the midguts were fixed in 4% paraformaldehyde in PBS for 1 h followed by incubation in permeabilization solution (0.1% Triton X-100) for 15 min. 100 µL of Label solution was used as a negative control. 50 µL of Enzyme solution was mixed with 450 µL of Label solution to form a TUNEL reaction mixture in which the midguts were incubated for 1 h at 37°C in the dark followed by washing thrice in PBS, and finally, the midguts were mounted in glass slides with DAPI to view under the microscope.

## Immunostaining analysis

Immunostaining for psyllid proteins as well as for Liberibacter was done as described previously (42). Briefly, the midguts were dissected out in PBS, fixed in 4% paraformaldehyde, treated with Triton-X for 30 min, and blocked with 1.5% bovine serum albumin for 1 h. The midguts were then incubated with primary antibody for Liberibacter with anti-OmpB- antibody (GenScript) (44), followed by secondary antibody conjugated with Cy3/Cy5 (Jackson ImmunoResearch Laboratories). This was followed by incubation with psyllid protein antibodies for Beclin1, anti-phospho Beclin1-Ser93/Ser96 (Cell signaling Technologies) for AMPK inhibition experiment or anti-phospho Beclin1-Thr119 (Sigma Aldrich, Israel) for DAPK inhibition experiment and a secondary antibody conjugated to

Cy3/Cy5. The midguts were washed at least thrice before mounting on a slide with DAPI and were visualized with Olympus IX81 confocal microscope.

## RNA isolation and qRT-PCR for gene expression

Total RNA was isolated from psyllid midguts and whole body from different experiments using Tri Reagent (Sigma Aldrich, Israel) as described previously. DNA contaminations were removed using DNaseI (Thermo Scientific). A minimum of 15 psyllids/midguts was used for each experiment. First-strand synthesis was carried out using M-MLV reverse transcriptase (Promega Corporations) following the manufacturer's instructions. Real-time PCR was carried out using ABsolute Blue SYBR green mix in a StepOne real-time PCR system (Applied Biosystems). Ct values were normalized using Elongation factor 1α (Ef1α) (45). The expression of each gene was calculated following Livak ($2^{-\Delta\Delta Ct}$) method (46) for relative gene expression.

## Quantification of CLso and qPCR

To quantify the relative abundance of *Ca*. Liberibacter solanacearum from each treatment, total DNA was isolated from individual psyllid/midgut from each experiment using modified CTAB protocol (47) as described in a previous study (42). Real-time analysis was carried out as described before using actin as a housekeeping gene. Each psyllid/midgut was crushed in 250 µL/100 µL of CTAB buffer, respectively, and were incubated for 1 h at 37°C followed by phenol-chloroform purification.

## Statistical analysis

The significance of relative expression analyses performed for both qRT-PCR and qPCR was determined using at least 10–12 samples each using one-way ANOVA with Tukey's post hoc test ($P < 0.05$).

## ACKNOWLEDGMENTS

We thank Eduard Belausov for technical help with the confocal microscope and members of the Ghanim laboratory for technical support and for providing comments on preliminary versions of the manuscript text.

This research was supported by grant 1163/18 from the Israel Science Foundation and grant 2019278 from US-Israel Binational Science Foundation to M.G.

## AUTHOR AFFILIATIONS

[1]Department of Entomology, Volcani Institute, Rishon LeZion, Israel
[2]Robert H. Smith Faculty of Agriculture, Food and Environment, Hebrew University of Jerusalem, Rehovot, Israel

## PRESENT ADDRESS

Poulami Sarkar, Department of Plant Pathology, Citrus Research and Education Center, Lake Alfred, Florida, USA

## AUTHOR ORCIDs

Poulami Sarkar  http://orcid.org/0000-0002-1822-3259
Ola Jassar  http://orcid.org/0000-0003-4618-1516
Murad Ghanim  http://orcid.org/0000-0001-6628-8308

## FUNDING

| Funder | Grant(s) | Author(s) |
|---|---|---|
| Israel Science Foundation (ISF) | 1163/18 | Murad Ghanim |
| United States - Israel Binational Science Foundation (BSF) | 2019278 | Murad Ghanim |

## AUTHOR CONTRIBUTIONS

Poulami Sarkar, Conceptualization, Data curation, Formal analysis, Investigation, Methodology, Validation, Visualization, Writing – original draft, Writing – review and editing | Ola Jassar, Formal analysis, Investigation, Methodology, Validation | Murad Ghanim, Conceptualization, Data curation, Funding acquisition, Project administration, Resources, Supervision, Writing – original draft, Writing – review and editing

## ADDITIONAL FILES

The following material is available online.

### Supplemental Material

**Supplemental figures (Spectrum01301-23-s0001.pdf).** Fig. S1 to S5.

### Open Peer Review

**PEER REVIEW HISTORY (review-history.pdf).** An accounting of the reviewer comments and feedback.

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
