## [Reviewer comments · Microbiology Spectrum]

Microbiology Spectrum

The plant pathogenic bacterium *Candidatus Liberibacter solanacearum* induces calcium-regulated autophagy in midgut cells of its insect vector *Bactericera trigonica*

Poulami Sarkar, Ola Jassar, and Murad Ghanim

Corresponding Author(s): Murad Ghanim, Agricultural Research Organization Volcani Center Information Center

Review Timeline:

Submission Date:	March 25, 2023
Editorial Decision:	June 2, 2023
Revision Received:	August 2, 2023
Accepted:	August 11, 2023

Editor: Joel Vega-Rodríguez

Reviewer(s): Disclosure of reviewer identity is with reference to reviewer comments included in decision letter(s). The following individuals involved in review of your submission have agreed to reveal their identity: Xin-Ru Wang (Reviewer #1)

Transaction Report:

DOI: <https://doi.org/10.1128/spectrum.01301-23>

June 2, 2023

Prof. Murad Ghanim
Agricultural Research Organization Volcani Center Information Center
Department of Entomology
Rishon LeZion 7505101
Israel

Re: Spectrum01301-23 (The plant pathogenic bacterium *Liberibacter solanacearum* induces calcium-regulated autophagy in midgut cells of its insect vector *Bactericera trigonica*)

Dear Prof. Murad Ghanim:

Thank you for submitting your manuscript to Microbiology Spectrum and apologies for the longer than expected review process. When submitting the revised version of your paper, please provide (1) point-by-point responses to the issues raised by the reviewers as file type "Response to Reviewers," not in your cover letter, and (2) a PDF file that indicates the changes from the original submission (by highlighting or underlining the changes) as file type "Marked Up Manuscript - For Review Only". Please use this link to submit your revised manuscript - we strongly recommend that you submit your paper within the next 60 days or reach out to me. Detailed instructions on submitting your revised paper are below.

Link Not Available

Sincerely,

Joel Vega-Rodríguez

Journals Department
Reviewer comments:

Reviewer #1 (Comments for the Author):

The manuscript explores the role of calcium ATPase, cytosolic calcium, and Beclin-1 in regulating autophagy and its association with *Liberibacter* in psyllids. The authors demonstrate the importance of these factors by silencing SERCA and ITPR genes and inhibiting Beclin1-phosphorylation through calcium-induced kinases. Based on their findings, they suggest a direct relationship between cytosolic calcium levels, autophagy, and CLso persistence and transmission in the carrot psyllid's midgut. Overall, the paper is well-written, and the experiments are sound. However, some further clarification or experimental procedures could strengthen the manuscript for publication.

1. To support the claim of increased ROS relative to CLso-, the authors may provide data on the intensity of the observation by

- quantifying the fluorescence. It would also be helpful if the authors could clarify how many fields or cells under confocal were quantitated by ImageJ.
2. For the silencing experiments, the authors need to provide additional information, such as the gene ID, any solutions used to evaluate gene target specificity, and clarify the knockdown efficiency.
 3. The authors may provide additional information to clarify why ATG2 was selected to determine this pathway and the expression patterns of other genes in this pathway. Furthermore, it would be interesting to know if inhibiting AMPK and DAPK can test the effects on autophagosomes/autolysosomes formation associated with Beclin-1.
 4. It would be helpful if the authors could provide any markers used to verify the autophagosomes in the psyllid midguts, such as monodansylcadaverine, since an increase in autolysosomes is not sufficient to conclude autophagy activation.
 5. To further investigate apoptosis, it would be beneficial if the authors could discuss additional markers in the discussion section, such as cleavage cas-3 and cytochrome c release, alongside the TUNEL assay employed to assess DNA fragmentation.
 6. The authors may provide information on the generation of psyllids used in this study.
 7. A scale bar could be added to Figure 4D.

Reviewer #3 (Comments for Author)

The manuscript submitted by Sarkar and Ghanim is well-written and technically sound. It provides a link between autophagy and persistence / transmission of *Liberibacter solanacearum* in its vector host, *Bactericera trigonica*, using fluorescence microscopy, RT-PCR and gene knockdown experiments. Some minor comments for consideration are provided:

Line 79 - unculturable does not account for the possibility that it will be cultured someday.

Line 165 - activstion should be activation

Line 225 - please indicate what primers were used for this in Table 1

Line 267 - please capitalize 'L' in *Liberibacter* throughout this section.

Line 271 - please provide the locus tag or protein ID for the ompB locus (I did not see this information in Ref 39).

Line 286 - please include the primers for Ef1a in Table 1.

Line 331 - italics missing throughout the references, several references are garbled and/or incomplete: 23, 29, 30, 31, 32, 36, 37, 39

Line 413 - Please state what the error bars in each graph represent. In most cases the 'n' is reported in the methods for these experiments, but it would be nice to have it included in the figure legend for the reader's benefit.

Staff Comments:

Preparing Revision Guidelines

Please return the manuscript within 60 days; if you cannot complete the modification within this time period, please contact me. If you do not wish to modify the manuscript and prefer to submit it to another journal, please notify me of your decision immediately so that the manuscript may be formally withdrawn from consideration by Microbiology Spectrum.

Response to Reviewer comments:

Reviewer 1:

1. To support the claim of increased ROS relative to CLso-, the authors may provide data on the intensity of the observation by quantifying the fluorescence. It would also be helpful if the authors could clarify how many fields or cells under confocal were quantitated by ImageJ.

Response: The intensities of the images are now depicted in supplementary Figure S1 with both the light microscopy images and the confocal images.

2. For the silencing experiments, the authors need to provide additional information, such as the gene ID, any solutions used to evaluate gene target specificity, and clarify the knockdown efficiency.

Response: The geneIDs have been now mentioned in Line 253-255 as well as in the footnotes of Table1. We checked the gene target specificity using qRT-PCR before carrying out further experiments. The protein expression was also validated using immunostaining to confirm the knockdown of the gene. This is now mentioned in Line 260-262.

3. The authors may provide additional information to clarify why ATG2 was selected to determine this pathway and the expression patterns of other genes in this pathway. Furthermore, it would be interesting to know if inhibiting AMPK and DAPK can test the effects on autophagosomes/autolysosomes formation associated with Beclin-1.

Response: This has now been explained in Line 195-199. Line 214-219 explains the effects of inhibiting AMPK and DAPK on Beclin phosphorylation and its effect on autophagy in the gut cells. We discovered that autophagy is mainly AMPK dependent pathway, and is independent of DAPK phosphorylation (Line 222-224).

4. It would be helpful if the authors could provide any markers used to verify the autophagosomes in the psyllid midguts, such as monodansylcadaverine, since an increase in autolysosomes is not sufficient to conclude autophagy activation.

Response: Thank you for the suggestion. We now used MDC stain after AMPK and DAPK inhibition, to observe the autophagic vacuoles. And the figure is available as supplementary Figure S5. (Line 153 & 157, and Line 277-281)

5. To further investigate apoptosis, it would be beneficial if the authors could discuss additional markers in the discussion section, such as cleavage cas-3 and cytochrome c release, alongside the TUNEL assay employed to assess DNA fragmentation.

Response: We lack sequence information for some of the apoptosis and autophagy specific genes and proteins for carrot psyllid, like specific caspases and LC3. We have discussed this in Line 207-209 and Line 224-229.

6. The authors may provide information on the generation of psyllids used in this study.

Response: The psyllid collection source has now been mentioned in Line 239.

7. A scale bar could be added to Figure 4D.

Response: The scale bar has been added now.

Reviewer 3:

1. Line 79 - unculturable does not account for the possibility that it will be cultured someday.

Response: Now Line 81, we changed to 'yet unculturable'

2. Line 165 - activstion should be activation

Response: Now in Line 169. Changed.

3. Line 225 - please indicate what primers were used for this in Table 1

Response: Reference added in Line 244.

4. Line 267 - please capitalize 'L' in Liberibacter throughout this section.

Response: Changed. Now Line 293

5. Line 271 - please provide the locus tag or protein ID for the ompB locus (I did not see this information in Ref 39).

Response: Now Line 297: The correct reference has now been added. Peptide OMP-B: "VIRRELGFSSEGDPIC"

6. Line 286 - please include the primers for Efla in Table 1.

Response: Now line 312. The reference has been added.

7. Line 331 - italics missing throughout the references, several references are garbled and/or incomplete: 23, 29, 30, 31, 32, 36, 37, 39

Response: The references have been corrected.

8. Line 413 - Please state what the error bars in each graph represent. In most cases the 'n' is reported in the methods for these experiments, but it would be nice to have it included in the figure legend for the reader's benefit.

Response: We included the error bars with n in each figure legend.

August 11, 2023

Prof. Murad Ghanim
Agricultural Research Organization Volcani Center Information Center
Department of Entomology
Rishon LeZion 7505101
Israel

Re: Spectrum01301-23R1 (The plant pathogenic bacterium *Candidatus Liberibacter solanacearum* induces calcium-regulated autophagy in midgut cells of its insect vector *Bactericera trigonica*)

Dear Prof. Murad Ghanim:

Your manuscript has been accepted, and I am forwarding it to the ASM Journals Department for publication. You will be notified when your proofs are ready to be viewed.

Sincerely,

Joel Vega-Rodríguez
Editor, Microbiology Spectrum
